# Benzo(a)pyrene and Cerium Dioxide Nanoparticles in Co-Exposure Impair Human Trophoblast Cell Stress Signaling

**DOI:** 10.3390/ijms24065439

**Published:** 2023-03-12

**Authors:** Gaëlle Deval, Margaux Nedder, Séverine Degrelle, Jasmina Rogozarski, Marie-Léone Vignaud, Audrey Chissey, Stacy Colzin, Christelle Laguillier-Morizot, Xavier Coumoul, Sonja Boland, Thierry Fournier, Amal Zerrad-Saadi, Ioana Ferecatu

**Affiliations:** 1Université Paris Cité, Inserm, 3PHM, F-75006 Paris, Francethierry.fournier@parisdescartes.fr (T.F.);; 2Inovarion, F-75005 Paris, France; 3Hormonology Department, Cochin Hospital, AP-HP, F-75014 Paris, France; 4Université Paris Cité, Inserm, UMR-S 1124, F-75006 Paris, France; 5Université Paris Cité, CNRS, Unité de Biologie Fonctionnelle et Adaptative, F-75013 Paris, France

**Keywords:** human placenta, chorionic villi, villous cytotrophoblasts, benzo(a)pyrene, cerium dioxide nanoparticles, AhR, CYP1A1, p53, p21, γ-H2AX

## Abstract

Human placenta is a multifunctional interface between maternal and fetal blood. Studying the impact of pollutants on this organ is crucial because many xenobiotics in maternal blood can accumulate in placental cells or pass into the fetal circulation. Benzo(a)pyrene (B*a*P) and cerium dioxide nanoparticles (CeO_2_ NP), which share the same emission sources, are found in ambient air pollution and also in maternal blood. The aim of the study was to depict the main signaling pathways modulated after exposure to B*a*P or CeO_2_ NP vs. co-exposure on both chorionic villi explants and villous cytotrophoblasts isolated from human term placenta. At nontoxic doses of pollutants, B*a*P is bioactivated by AhR xenobiotic metabolizing enzymes, leading to DNA damage with an increase in γ-H2AX, the stabilization of stress transcription factor p53, and the induction of its target p21. These effects are reproduced in co-exposure with CeO_2_ NP, except for the increase in γ-H2AX, which suggests a modulation of the genotoxic effect of B*a*P by CeO_2_ NP. Moreover, CeO_2_ NP in individual and co-exposure lead to a decrease in Prx-SO_3_, suggesting an antioxidant effect. This study is the first to identify the signaling pathways modulated after co-exposure to these two pollutants, which are common in the environment.

## 1. Introduction

The human placenta is a transitory organ at the interface between the maternal and fetal circulation. This organ ensures essential functions for the maintenance of pregnancy and for fetal development [1]. The placenta provides nutrient and gaseous transport, elimination of waste products, and intensive endocrine function by producing several hormones (human chorionic gonadotropin (hCG) and steroid hormones) essential for placental development and metabolism. The placenta is also a selective barrier that assures protection of the fetus against the many xenobiotics to which the mother is exposed. However, many substances can accumulate in and/or cross the placenta via different mechanisms and consequently reach the fetus. The placenta is composed of chorionic villi, which are the structural and functional units of the human placenta. These villi consist of a mesenchymal core in which the fetal vessels are embedded and which is bordered by an epithelium. This epithelium is formed by the outer syncytiotrophoblast (ST) layer, a multinucleated syncytium, and the underlying mononuclear villous cytotrophoblasts (VCT), which constantly differentiate and fuse to renew the syncytium. VCT can be purified from human placenta obtained following a term delivery. When cultivated in vitro, these cells retain the ability to aggregate and fuse spontaneously after 24–48 h to form the ST and are, therefore, a better model of physiological conditions than typical trophoblast cell lines, such as the BeWo choriocarcinoma trophoblasts. From 10 gestational weeks, the placental chorionic villi bathe freely in maternal blood in the intervillous space. Therefore, any foreign molecules with a functional activity, such as pollutants that may accumulate within the placenta, can directly impact placental functioning, with repercussions on pregnancy outcome [2].

Environmental pollutants such as polycyclic aromatic hydrocarbon (PAH) and nanoparticles (particles whose three dimensions are under 100 nm) are widespread contaminants. One of the most represented PAH is benzo(a)pyrene (B*a*P), a five-ring benzene that has been classified by the International Agency for Research on Cancer in group I (a carcinogenic, mutagenic, and reprotoxic agent for humans) and which is also an endocrine disruptor [3]. B*a*P is formed by incomplete combustion of organic compounds and is found in cigarette smoke, industrial waste, charbroiled food (e.g., meat, fish, vegetables, bread, etc.), and in car exhaust fumes (diesel). One of the main cellular targets of B*a*P is the ligand-activated transcription factor AhR: the aryl hydrocarbon receptor, which is involved in xenobiotic detoxification by inducing phase I and phase II detoxifying enzymes [4]. The induction of these enzymes allows the formation of B*a*P metabolites, the majority of which are eliminated from the body through phase III membrane transporters. However, in the case of PAH and specifically B*a*P, such processes may also contribute to B*a*P mutagenesis and toxicity. Indeed, the AhR-dependent metabolization of B*a*P may lead to reactive electrophilic metabolites (such as benzo-(a)-pyrene-diol epoxides) that form adducts with DNA. Moreover, phase I enzymes may also generate reactive oxygen species (ROS) through decoupling enzymatic events. In human placenta, AhR is involved in biological responses to B*a*P (detoxification but also bioactivation). We previously showed that B*a*P induces phase I genes (CYP1A1, CYP1B1, and CYP1A2) in human primary trophoblasts [5]. Several cohort studies found PAH (including B*a*P) in maternal blood [6], placenta, cord, and milk [7] from both smoking and nonsmoking women. In women that smoked during pregnancy, B*a*P was detected in both placenta and in umbilical cord blood, together with B*a*P–DNA adducts [8,9]. During pregnancy, exposure to B*a*P is correlated with an increased risk of early miscarriage [10] and complications of pregnancy such as prematurity [11] and lower birth weight [12].

Cerium (Ce) is a metal of the lanthanide group, which oxidizes rapidly on contact with air. It has two ionizing oxidation states: Ce^3+^ and Ce^4+^. Cerium dioxide nanoparticles (CeO_2_ NP) gained the attention of the scientific community because they have been used in numerous applications since the 2000s [13], for example, as additives in diesel fuels (EnvirOx) and in cigarettes, and as polishing agents (glass, ceramic). These nanoparticles are also studied as potential medicines (e.g., antitumor drugs) [14,15] for their pro- and antioxidant properties, which depend on multiple factors, such as the Ce^3+^/Ce^4+^ ratio, dose, and exposure time. When Ce is added to diesel fuels, particles released into the environment by cars [16,17] have a 6.5% increased Ce content [14,18]. In 2010, the Organization of Economic Cooperation and Development included CeO_2_ NP in the top priority list of nanomaterials requiring urgent evaluation. There are few data concerning the impact of CeO_2_ NP, to which pregnant women may be involuntarily exposed. Recent publications reported the detection of Ce in maternal blood, urine, and breast milk, but these data remain scarce, with significant differences depending on the place of residence of the pregnant women (from 21 to 70 ng/L in serum samples from Spanish women in 2009 and from 10 to 5180 ng/L in serum samples from pregnant women in Shanxi Province, China, between 2010 and 2018) [19,20]. An increased concentration of cerium in maternal blood and urine has been associated with a risk of fetal neural tube defects and a reduced level of thyroid-stimulating hormone in newborns [19,21]. CeO_2_ NP also appear to be responsible for placentation abnormalities [22], in particular by the excessive activation of autophagy and a decrease in invasive and migratory abilities in first-trimester HTR-8/SVneo trophoblast cell lines [23,24]. We previously showed that CeO_2_ NP have adverse effects on term human primary trophoblasts, affecting their viability in a dose-dependent manner, their capacity to differentiate to form ST, and the endocrine function of the placenta, with disruption of the secretion of peptides, such as hCG and steroid hormones [25]. However, humans are rarely exposed to CeO_2_ NP alone. As a result of their high surface-to-volume ratio and high reactivity, CeO_2_ NP can interact with other pollutants from common sources, including fuel-borne chemicals such as B*a*P, the emission of which is increased by 35% in cerium-additive fuels [17,18,26]. Hence, CeO_2_ NP and B*a*P are encountered in combination (from common combustion processes such as diesel exhaust, cigarette smoke, etc.) and human exposure to both contaminants in a mixture is common. Prior exposure to NP can alter the barrier and defense functions of trophoblasts (metabolism, endocrine function, antioxidant defense, signaling pathways, etc.), which could become more vulnerable to other chemical pollutants (such as B*a*P). In addition, the lipophilic properties of B*a*P may modify the mechanism and kinetics of NP cellular uptake. Human exposure to these two pollutants is mainly through the respiratory tract. Subsequently, lipophilic B*a*P and NP, including CeO_2_ NP, are able to cross biological barriers, enter the systemic circulation, and reach blood-irrigated organs, such as the placenta [27,28]. As compared with individual exposure, these pollutants in combination may synergistically affect the proper functioning of the placenta by activating different cellular pathways in parallel.

In this study, we investigated the molecular consequences of co-exposure to CeO_2_ NP and B*a*P vs. their individual exposure in terms of the major actors of cellular stress by using chorionic villi and primary cytotrophoblasts purified from human placentas at term of pregnancy. The ratio of pollutants tested here mimics the airborne levels of PAH found on atmospheric ultrafine particles (UFP) in Paris [29] (from 0.15 to 2 mg PAH per g of UFP), which we adjusted to 1 µM of B*a*P per 10 µg/cm^2^ of CeO_2_ NP in our study. Here, we show that B*a*P activates the AhR signaling pathway (measured by CYP1A1 induction) and DNA damage, with the stabilization of p53, its target p21, and phosphorylated H2AX (ɣ-H2AX), but without the activation of the antioxidant defense system. Both AhR antagonist and p53 inhibitor diminished p21 induction, although p53 inhibition had no effect on DNA damage. Furthermore, co-exposure of trophoblasts to B*a*P and CeO_2_ did not modify B*a*P metabolization and p53 activation, suggesting that B*a*P has a dominant effect. However, no DNA damage was observed after co-exposure, potentially due to the antioxidant effect of CeO_2_ NP suggested by a decrease in Prx-SO_3_.

## 2. Results

### 2.1. Activation of the AhR Signaling Pathway by BaP in Human Primary VCT

We first determined the effect of B*a*P on VCT viability, as we already reported data for CeO_2_ NP in our previous study [25]. For this, VCT were exposed to a range of B*a*P concentrations from 0.1 to 8 μM for 24 h to 72 h (maximum culture time of primary VCT). WST-1 colorimetric metabolic activity assays were performed after each incubation period (Figure 1A). A high concentration of B*a*P (above 2 μM) decreased trophoblast metabolic activity only after a 72 h of exposure and there was no statistical difference at shorter periods (24 h or 48 h). In our previously published study using BeWo trophoblast cell lines, we showed that 1 µM of B*a*P activates AhR, resulting in increased CYP1A1 expression [5]. Thus, we assessed whether lower concentrations of B*a*P (until 0.1 μM) and incubation times (from 30 min) could still induce the AhR-dependent bioactivation pathway, which could be more representative of the placental environment. Moreover, as trophoblast cell lines can react differently to pollutants, we assessed whether the activation of the AhR pathway was reproduced in primary VCT isolated from term human placentas. We confirmed the absence of apoptosis induction by B*a*P at these concentrations by Western blot by following the cleavage of the apoptotic marker PARP-1, a target of caspases (Figure 1B). PARP-1, a nuclear poly (ADP-ribose) polymerase involved in DNA repair, is a caspase-3 direct target, whose cleavage (cleaved PARP-1) serves as a marker of cells undergoing apoptosis. Staurosporine (STS), a classic apoptosis inducer, clearly led to the cleavage of PARP-1 (cleaved PARP-1, 89 kDa vs. uncleaved PARP-1, 116 kDa) from 2 h of exposure (Figure 1B). In comparison, the exposure to increasing concentrations of B*a*P did not induce any cleavage of PARP-1 for any of the exposure times (30 min and/or 24 h). We then verified the activation of B*a*P metabolism according to the different concentrations (0.1, 0.6, and 1 μM) and incubation times (30 min and 24 h) in VCT (Figure 1B) by monitoring the protein levels of AhR and its main transcriptional target CYP1A1 by Western blot. Generally, AhR is not modulated at the protein level and its genomic activity is linked to its nuclear translocation after ligand binding and induction of CYP1A1. However, we previously demonstrated a constitutive nuclear localization of AhR in human primary VCT, in the absence or presence of B*a*P [5]. The protein level of AhR, corresponding to the two bands (72 and 95 kDa, previously validated [5]), varied slightly over the 24 h of treatment. In parallel, the level of CYP1A1 increased in a time-shifted manner (after 24 h) and in a concentration-dependent manner from 0.1 µM of B*a*P (Figure 1B). The use of an AhR antagonist (CH223191) led to a 50% decrease in the CYP1A1 protein level after B*a*P treatment (0.6 µM), corroborating AhR-dependent activation (Appendix A). Thus, although B*a*P did not appear to be cytotoxic for VCT, it was able to activate the AhR pathway from 0.1 μM in primary trophoblasts, reproducing the cellular effects previously obtained with 1 μM of B*a*P on the BeWo cell line [5].

### 2.2. Trophoblast Exposure to BaP and CeO_2_

Thereafter, in order to study the effects of co-exposure to B*a*P and CeO_2_ NP, we carried out a kinetic study (30 min, 4 h, and 24 h) for both pollutants, i.e., B*a*P and CeO_2_ NP, in individual exposures. In order to mimic the airborne levels of these pollutants when they are encountered together (from 0.15 to 2 mg PAH per g of UFP), the concentrations tested correspond to 6.3 µg/cm^2^ of CeO_2_ NP (40 µg/mL), which was nontoxic in our previous publication [25], and 0.6 µM of B*a*P. We followed the protein levels by Western blot of AhR, its target CYP1A1, and PARP-1. As shown in Figure 2, the protein levels of CYP1A1 increased in a time-dependent manner after exposure to B*a*P, while AhR varied slightly over the 24 h of treatment (Figure 2A). For CeO_2_ NP, there was no major variation in the rate of CYP1A1. STS treatment led to the cleavage of PARP-1, but, again, this was not the case after exposure to either B*a*P or CeO_2_ NP, indicating that the AhR pathway is only induced with B*a*P (as expected) without the induction of apoptosis in VCT.

We also determined the cytotoxicity of co-exposure to B*a*P and CeO_2_ NP in human primary trophoblasts by WST-1. In order to do so, VCT were exposed to a range of concentrations of B*a*P and CeO_2_ NP starting from 12.6 µg/cm^2^ of CeO_2_ NP (80 µg/mL) and 1.28 µM of B*a*P (respecting the ratio of the two pollutants from the atmospheric environmental data) for 72 h, making serial two-fold dilutions until 0.4 µg/cm^2^ of CeO_2_ NP and 0.04 µM of B*a*P to be close to the placental environmental exposure. Since B*a*P is not cytotoxic at 1 µM for a 72 h incubation time, we compared co-exposure with B*a*P to exposure to CeO_2_ NP alone (Figure 1A). There was a significant decrease in the metabolic activity of VCT at the highest doses of CeO_2_ NP and after co-exposure with B*a*P. However, there was no significant difference between co-exposure and NP alone (Figure 2B). We also assessed whether the ratio of 1 µM of B*a*P to 10 µg/cm^2^ of CeO_2_ NP modified hCG secretion. Since hCG is mainly produced by the ST, we assayed the hCG after 48 h of treatment so that the VCT had time to differentiate into ST. The results showed no significant change in hCG secretion after treatment compared to the control (Figure 2C).

### 2.3. Analysis of Activated Signaling Pathways by BaP and CeO_2_ NP Using Cell Stress Array

In order to determine which protein levels of major cellular stress factors are modulated after exposure of primary VCT to BaP and NP CeO2, alone and in co-exposure, we carried out a small proteomic analysis by the Cell Stress Array. This assay is a sensitive analysis for the determination of the relative levels of 26 selected human cell stress-related proteins. VCT from five independent placentas were incubated with B*a*P and/or CeO_2_ NP at the concentration and time used previously (0.6 µM of B*a*P and 6.3 µg/cm^2^ of CeO_2_ NP) in individual exposure or co-exposure for 24 h. Before performing the Cell Stress Array, we wanted to make sure that the VCT response from these five individual placentas was consistent. In order to do so, we selected the AhR signaling pathway as the biomarker pathway, due to its particular sensitivity regarding B*a*P exposure, and measured the protein level of CYP1A1 by Western blot (Figure 3 shows the mean of *n* = 5 and Appendix A shows the five independent placentas). The statistical analysis shows a significant increase in the level of CYP1A1 (seven times the control level) after B*a*P alone or in co-exposure with CeO_2_ NP. No difference was detected between the two conditions (B*a*P vs. B*a*P + CeO_2_). VCT cultures from different placentas responded in the same way toward the AhR pathway after incubation with these pollutants and, therefore, a mixture of placental protein samples from these five placentas could be used as representative for the Cell Stress Array when added to a proteome chip for each condition.

Thereafter, we mixed equal quantities of the five different placental total protein extracts of VCT for each condition (dimethyl sulfoxide (DMSO) as control, B*a*P, CeO_2_, and B*a*P + CeO_2_) in order to test a representative sample on membranes from the Cell Stress Array study. On the resulted blots, for each spot the signal intensity was normalized with respect to the DMSO control condition (Appendix A). The data are represented as a heatmap with a color gradient for a difference of 10% (Figure 4). After individual exposure to B*a*P, three factors were increased: we noted a coherent increase in phospho-p53 Ser 46 (+15.6%), its transcriptional target p21 (+41.4%), and the metalloproteinase ADAMTS1 (+10.2%). Among the protein levels that decreased as a result of B*a*P, we noted major decreases in hypoxia-inducible factor HIF-2α (−30.5%), heat shock protein HSP60 (−19.8%), phospho-JNK (−14.3%), phospho-p38 (−10.3%), and PON1 (−10.4%). After exposure to CeO_2_ NP alone, we found that many more protein levels decreased as compared to exposure to B*a*P alone, and none increased. We noted a decrease in the levels of chaperones such as HSP70 (−28.9%), phospho-HSP27, and phospho-p38 (−25.1% and −21%, respectively). Similar to B*a*P, CeO_2_ treatment decreased the levels of HIF-2α (−26.7%), but also decreased HIF-1α levels and its transcriptional target carbonic anhydrase (CA9, −12.6%). Contrary to B*a*P, after VCT exposure to CeO_2_ NP alone, there was no phosphorylation of p53 and there was a decrease in its target p21 (−11.1%) and the anti-apoptotic protein Bcl2 (−11.2%). After VCT co-exposures to B*a*P and CeO_2_ NP, overall, there was an accentuation of the decrease observed in individual exposures. Indeed, eight major stress actors were reduced in terms of protein level by co-exposure to the two pollutants, including hypoxia-inducible factor HIF-2α (−47.9%), HIF-1α (−28.8%) and its main target gene CA9 (−22%), inflammation factor NF-κB1 (−29.2%), redox SIRT2 (−27.4%) and DKK4 (−20.5%). Only the p21 protein increased compared to the DMSO control (by 30.3%) but decreased when compared to B*a*P alone (−11.1%). Interestingly, three new proteins were reduced by co-exposure to B*a*P and CeO_2_: PON3 (−18.9%), Mn SOD2 (−13.7%), and p27 (−13%), but were unmodified by VCT exposure to each pollutant individually.

### 2.4. Oxidative Status of Villous Cytotrophoblasts

Using Western blot, we then investigated some of the main factors influencing redox status, since the levels of hypoxia and inflammation factors varied in the Cell Stress Array. We used protein extracts from the same term placentas used for the Cell Stress Array that were treated with B*a*P and/or CeO_2_. We evaluated the protein levels of Prx-SO_3_, subunit NFκB p65, and HO-1 (Figure 5). *Tert*-butyl hydroperoxide (TBHP) is described as a strong oxidative stress inducer and was used here as a positive control to show that trophoblasts are sensitive to H_2_O_2_, as reported before [25,30], and to STS as an apoptosis inducer. As shown in the graph in Figure 5, the overoxidation of peroxiredoxin (Prx-SO_3_) decreased significantly in the presence of CeO_2_ NP alone or after co-exposure with B*a*P, suggesting that these NP have an antioxidant effect, while the protein levels of HO-1 and NFκB p65 did not vary, in contrast to what was observed for NFκB1 (p50) in the Cell Stress Array.

### 2.5. The p53 Signaling Pathway Is Induced by BaP in Primary Human Trophoblasts

Subsequently, we explored the p53 signaling pathway and its transcriptional target p21 by Western blot and real-time quantitative PCR (RT-qPCR) in trophoblasts and directly via immunohistochemistry (IHC) in whole chorionic villi tissues. We also assessed whether the p53 signaling pathway depends on the activation of the AhR pathway. We used protein and RNA extracts from VCT that were treated with B*a*P and/or CeO_2_ NP at 1 µM of B*a*P and 10 µg/cm^2^ of CeO_2_ NP for 24 h (Figure 6). As seen in Figure 3, at lower doses, 1 μM of B*a*P in individual exposure or in co-exposure with CeO_2_ NP activates the AhR pathway with the induction of CYP1A1 (Figure 6A). The relative amount of CYP1A1 mRNA was increased 72- and 68-fold for individual exposure and co-exposure, respectively (Figure 6B). Once again, CeO_2_ NP in individual exposure did not induce any modification of CYP1A1 at the transcript or protein level. For individual exposure to B*a*P, the relative amounts of p53 protein and mRNA significantly increased (2.4- and 2-fold, respectively), as did the protein and mRNA levels of its target p21 (1.9- and 3.5-fold, respectively, vs. control), suggesting p53 activation (Figure 6A,B). Exposure to CeO_2_ NP alone did not stabilize p53 or induce p21 at the protein and transcript levels (Figure 6A,B). For the co-exposure to B*a*P and CeO_2_, the protein levels for both p53 and p21 were significantly higher than the control (2.2- and 1.6- fold, respectively). This was reproduced at the mRNA level for p21 (a 4.4-fold change vs. control). Interestingly, B*a*P in individual exposure significantly increased the level of ɣ-H2AX (1.6-fold vs. control), indicating double-stranded DNA damage. However, co-exposure to B*a*P with CeO_2_ NP reduced the level of ɣ-H2AX to the control level, suggesting that CeO_2_ has antioxidant activity (Figure 6A and quantification plot).

These results confirm those obtained by the Cell Stress Array. We next investigated the signaling pathways of B*a*P metabolization and p53 using an AhR pathway antagonist (CH-223191) and pifithrin-α, an inhibitor of the transactivation of p53-responsive genes. We used protein and mRNA extracts from VCT pretreated with the two inhibitors and/or 1 µM of B*a*P for 24 h (Figure 7). The effectiveness of the AhR antagonist CH-223191 was confirmed by the fact that the CYP1A1 protein level decreased significantly from 7.7- to 3.4-fold the control level for the individual exposure to B*a*P vs. B*a*P with AhR antagonist (Figure 7A). When VCT were exposed to B*a*P, the relative amounts of the p53, p21, and ɣ-H2AX proteins were significantly lower in the presence of the AhR antagonist (1.6-, 1.2-, and 0.9-fold vs. 2.6-, 2-, and 1.6-fold, respectively, Figure 7B). These results indicate that the stabilization of p53 and the subsequent induction of p21 together with B*a*P-induced double-stranded DNA damage are AhR-dependent. When VCT were exposed to B*a*P in the presence of the p53-dependent inhibitor pifithrin-α, the relative amount of p21 was reduced to the DMSO level. (Figure 7C). These results confirmed that the induction of p21 depends on the promotion of p53 transcriptional activities. An analysis of the relative quantity of mRNA p21 under the same conditions corroborated the results obtained in the Western blot, with a 3.5-fold increase after exposure to B*a*P vs. a 1.8-fold increase on the addition of the AhR antagonist and a 1.3-fold increase on the addition of pifithrin-α (Figure 7D).

### 2.6. p53 and HIF Signaling in Chorionic Villi Explants Exposed to BaP and/or CeO_2_ NP

Because the impact of B*a*P and CeO_2_ NP was particularly remarkable in the Cell Stress Array for HIF, p53, and p21, we assessed whether the effects observed previously were reproduced by using a different model of VCT primary cultures. Thus, on chorionic villi explants exposed to these pollutants, we quantified the presence of these proteins after IHC (Figure 8). In all villi cells, B*a*P significantly decreased both HIF-1α and HIF-2α, whereas B*a*P and CeO_2_ NP co-exposure significantly decreased HIF-2α, as did CeO_2_ NP, thus corroborating the results obtained on VCT by the Cell Stress Array. Moreover, the stabilization of p53 followed by the induction of p21 by both B*a*P and co-exposure with CeO_2_ NP was also reproduced in the exposed chorionic villi explants. However, contrary to the results obtained on trophoblasts by the Cell Stress Array, Western blot, and RT-qPCR, exposure of villi explants to CeO_2_ NP alone also led to p53 stabilization and the induction of its transcriptional target p21. This explant model allows for a more realistic exposure to contaminants: by exposing the ST layer to direct contact with pollutants (rather than VCT), it reproduces the dynamics of villi contact with maternal blood as source of pollutants in the intervillous chamber, . It also highlights the impact of pollutants on all cell types present in the chorionic villi, not only on the trophoblast layer.

## 3. Discussion

In this study, we aimed to determine the major factors in cellular stress, i.e., the factors whose expression is modulated after individual exposure and co-exposure of primary human VCT to two major environmental pollutants: B*a*P and CeO_2_ NP. Trophoblastic cell lines (such as BeWo, JEG, JAR) represent one of the most used trophoblastic models in placental toxicology, but they are either modified or derived from choriocarcinoma cells with an indefinite proliferation phenotype. BeWo cells also require the addition of forskolin (an activator of PKA via cyclic AMP) in order to terminally differentiate to form the ST. For all these reasons, we used primary VCT purified from term human placenta because they are a better model of physiological conditions than trophoblast cell lines due to their capacity to spontaneously aggregate and fuse in vitro at 24–48 h of culture to form the ST. Primary VCT are nonproliferating cells in vitro.

Since the exact B*a*P/CeO_2_ NP ratio in ambient air or in maternal blood is not known and is difficult to measure due to large variations depending on place of residence, lifestyle (diet, smoking), etc., the ratio of pollutants tested here is close to the airborne levels of PAH found on UFP in Paris [31] (from 0.15 to 2 ng PAH per µg of UFP). We used 0.6 µM of B*a*P per 6.3 µg/cm^2^ of CeO_2_ NP in the Cell Stress Array and we used the same ratio but rounded up to 1 µM per 10 µg/cm^2^ in the other studies. Because the molar mass of B*a*P is 252.3 g/mol, the ratio of 1 µM of B*a*P (252.3 µg/L) per 10 µg/cm^2^ of CeO^2^ NP corresponds to 6.63 ng of B*a*P per µg of CeO_2_ NP. We showed in our previous publication that CeO_2_ NP concentrations below 80 µg/mL (12.6 µg/cm^2^) did not affect the metabolic activity of VCT at the exposure time chosen for our experiments (24 h), but did disrupt both trophoblast morphological and functional differentiation [25].

We show here that B*a*P only affected the metabolic activity of primary trophoblasts at concentrations higher than 2 μM and after prolonged exposure (72 h) (Figure 1A). The authors and others showed that exposure to B*a*P leads to the activation of the AhR pathway and increased expression of CYP1A1 and CYP1B1, which are involved in the bioactivation of B*a*P, including in the BeWo cell line [5,32]. We showed the activation of the AhR signaling pathways by B*a*P via a dose-dependent (0.1 to 1 μM) increase in protein expression of CYP1A1 after 24 h of exposure (Figure 1C), with the lowest concentration of 0.1 µM approaching those found in maternal blood. Then, we confirmed that lower concentrations of B*a*P (0.1, 0.6 μM), which are more representative of the placenta, had cellular effects similar to those of 1 μM of B*a*P. Our results for CYP1A1 converge towards the activation of the detoxification pathway after exposure to B*a*P. This is emphasized by the use of the AhR antagonist, CH223191, which partially inhibited the induction of CYP1A1 after B*a*P incubation (Appendix A).

In order to identify other signaling pathways modulated by exposure to B*a*P and/or CeO_2_ NP, we used the Cell Stress Array to observe fine changes in the levels of the main cellular stress proteins and factors. This test was carried out at nontoxic doses of pollutants and after 24 h of exposure in order to be sure that the AhR pathway had, indeed, been activated at this stage. However, the chosen dose of NP used here (6.3 µg/cm^2^ representing 40 µg/mL) was able to disturb placental functions (morphological and functional differentiation), as reported in our previous publication [25]. Both the WST-1 cell viability assay and the absence of cleavage of PARP-1, a target of caspase 3, highlight that B*a*P does not induce apoptosis at the tested concentration and times (0.6 μM for 30 min, 4 h, and 24 h) (Figure 2B).

Concerning CeO_2_ NP, we found a significant decrease in the superoxidized form of peroxiredoxins (Prx-SO_3_) compared to the control after 24 h of individual and co-exposure to CeO_2_ NP in the five placentas (Figure 5). Peroxiredoxins are proteins with intracellular H_2_O_2_ sensor thiol groups. Their superoxidized form Prx-SO_3_ (excessive oxidation) indicates an abundance of hydrogen peroxide. This decrease in Prx-SO_3_ after exposure to CeO_2_ NP and co-exposure suggests that these nanoparticles have an antioxidant effect.

Furthermore, an analysis of the results of the Cell Stress Array also showed a decrease in Mn-SOD2 with the co-exposure to CeO_2_ NP and B*a*P. CeO_2_ NP notably induced a reduction in the proteins involved in the redox-sensitive pathway, such as p38/HSP27 (−21% and −25%, respectively) and those involving HSP70 (−29%). The protein p38, from the MAPK family (mitogen-activated protein kinase), is involved in different signaling pathways, including the cell cycle, differentiation, cell stress, and cell death. After phosphorylation under oxidative stress, p38 activates a MAPKAP-K2 (protein kinase-2 activated by MAP kinase 2) involved in the change in conformation by phosphorylation of the heat shock protein HSP27. Phosphorylated HSP27 is involved in the remodeling of the actin network in particular. Microvilli containing actin are present on the apical membrane of ST and their remodeling processes are essential for trophoblast differentiation and function [33]. These decreases in p38 and HSP27 protein levels can be correlated with the results obtained previously by our laboratory showing the decrease in the fusion of VCT into the ST after exposure to CeO_2_ NP at the same concentration [25]. The protein expression of HSP70 can be modulated by different mechanisms under conditions of oxidative stress. Caution is needed regarding our interpretations because the results of the Cell Stress Array were obtained at a single exposure time of 24 h. It would, therefore, be interesting to study the p38 pathway at very short exposure times (15 min, 30 min, and 1 h) to CeO_2_ NP alone in order to discover the pathways involved in the variation in phosphorylation level.

NFκB in its dimer form (p50/p65) is a transcription factor activated by oxidative stress. Under cellular stress, NFκB is translocated in the nucleus and participates in the process of cytokine secretion, angiogenesis, and the inhibition of apoptosis. However, the Cell Stress Array analysis showed a decrease in NFκB1 subunit (p50) levels and an absence of significant modification of the protein level of the NFκB p65 subunit by Western blot when exposed to CeO_2_ NP alone or in co-exposure with B*a*P (Figure 5). However, we know that the activity of NFκB is mainly regulated at the level of its nuclear localization. Studying the translocation of NFκB by IHC could provide more information.

In summary, CeO_2_ NP have an antioxidant potential in individual exposure or in co-exposure with B*a*P. Our observations of these NP are in agreement with those of our previous report [25] and with other data from the literature [34], describing an antioxidant effect probably by acting as a ROS scavenger, with a mimetic activity of catalase at the NP surface created by oxygen vacancies [35].

Both the Cell Stress Array and IHC revealed that exposure to CeO_2_ NP led to a decrease in the protein levels of HIF-2α, with a greater decrease in co-exposure in the Cell Stress Array, which suggests a synergistic effect. However, different results were seen for HIF-1α in the Cell Stress Array and IHC after exposure to B*a*P and/or CeO_2_ NP. Our attempt at validating this via Western blot was unsuccessful because we did not have quantifiable HIF bands without hypoxic chemical induction (such as cobalt chloride). Nevertheless, HIF-1α stabilization by CeO_2_ NP was reported by Das et al., showing that a high Ce^3+^/Ce^4+^ ratio can activate angiogenesis via HIF-1α stabilization in HUVEC cells [36]. In addition, the metalloproteinase ADAMTS1, which was increased by exposure to B*a*P in the Cell Stress Array (Figure 4), is induced by hypoxia in HUVEC cells by binding of HIF-1α to the promoter of ADAMTS1. The first stages of placental development in the first trimester take place in a low O_2_ environment (1–2%). HIF are stabilized during this critical period and play essential roles in placental morphogenesis, trophoblast proliferation, and angiogenesis [37]. These roles have been described using gene knockout mouse models. HIF-1α^−/−^ and HIF-2α^−/−^ embryos present placental vascularization abnormalities [38]. However, in the term placenta, the oxygen environment is approximately 8% and HIF are degraded by the proteasome. High levels of HIF-1α in term placentas are associated with placental pathologies such as preeclampsia and intrauterine growth retardation [39]. A large number of studies show that HIF-1α and HIF-2α do not have the same expression profile during hypoxia. HIF-1α is activated early but for a limited time while HIF-2α is activated later but for longer. Recent transcriptome studies have shown different target gene panels depending on whether HIF-1α or HIF-2α is activated by hypoxia [40]. HIF targets include VEGF, sFlt-1, and angiotensin 2, which have roles in angiogenesis [40]. It has been demonstrated that HIF-1α/ARNT dimers compete for the ARNT bond [41]. However, we noted a temporary increase in AhR after exposure to B*a*P for 30 min. AhR could, therefore, compete with HIF-1α. In view of the modifications observed in the levels of HIF in term VCT, additional studies should be carried out in this particular pathway in order to determine whether chronic exposure to B*a*P and/or CeO_2_ NP in the first trimester of pregnancy induces angiogenesis abnormalities by changes in HIF levels.

Regarding the pro-oxidant effect of B*a*P, exposure to 0.6 μM of B*a*P for 24 h did not induce an increase in the protein level of Prx-SO_3_ according to Western blot (Figure 5). We studied the second line of defense against oxidative stress by analyzing protein levels of heme oxygenase-1 (HO-1) by Western blot, but we did not observe an increase in HO-1 levels after 24 h of exposure to B*a*P (Figure 5). These results combined with those of the Cell Stress Array for individual exposure to B*a*P led to the conclusion that, at a concentration of 0.6 µM and 24 h of exposure, B*a*P does not modulate the protein expression of antioxidant defenses.

However, the Cell Stress Array revealed an increase in the phosphorylation of p53 on serine 46 after exposure to 0.6 µM of B*a*P (Figure 4). These results were confirmed by Western blot after 24 h of the 1 µM B*a*P treatment (Figure 6). In the absence of cellular stress, p53 is localized in the cytoplasm and is maintained at a low level of expression by a negative feedback loop mediated by an E3 ubiquitin ligase such as Mdm2, leading to degradation by the proteasome. The p53 protein is stabilized by phosphorylation at multiple sites (such as serine 15 or serine 46) under conditions of cellular stress, such as oxidative stress, DNA damage, and hypoxia [42]. B*a*P at the times and doses tested did not lead to overoxidation of Prx-SO_3_. However, Western blotting revealed an increase in γ-H2AX, the phosphorylated form of the histone H2AX, signaling double-stranded DNA damage after B*a*P treatment (Figure 6). The accumulation of phosphorylated p53 in the cytoplasm leads to its nuclear translocation for the activation of target genes, which leads to different cellular fates (cell cycle arrest, quiescence, senescence, or apoptosis). The increase in the protein level of p53 does not seem to be linked to the induction of apoptosis in trophoblasts incubated with B*a*P (Figure 1). This could instead be linked to the genotoxic effect of B*a*P with the increase in γ-H2AX after B*a*P treatment (Figure 6). Interestingly, co-exposure did not lead to an increase in γ-H2AX compared to the control. This difference between exposure to B*a*P alone and co-exposure could be due to the potential antioxidant effect of CeO_2_ NP, counterbalancing the genotoxic effect of B*a*P. Indeed, CeO_2_ NP are currently being studied for therapeutic applications due to their antioxidant properties and their ability to reduce DNA damage [43].

In parallel, the Cell Stress Array shows a 41% increase in the p21 protein level after exposure to B*a*P and a 30% increase after co-exposure to B*a*P and CeO_2_ NP compared to the control. In addition, p21, which is also known as cyclin-dependent kinase inhibitor 1, is a transcriptional target of p53 involved in cell cycle regulation and senescence. This result was then confirmed by Western blotting and RT-qPCR (Figure 6). We confirmed that the induction of p21 after treatment with B*a*P was due to the action of the transcription factor p53 using pifithrin-α, an inhibitor of the transactivation of p53-responsive genes. Moreover, pifithrin-α is also a potent agonist of AhR [44]. The use of this inhibitor with B*a*P led to p21 mRNA and protein levels close to control values (Figure 7C). These observations are in agreement with those reported in the literature for trophoblastic cell lines. For example, in the JEG-3 choriocarcinoma cell line, exposure to B*a*P leads to DNA damage with the phosphorylation of p53 on serine 15 and the induction of the expression of p21, which is responsible for cycle cell arrest in the G2/M phase, without leading to cell apoptosis [45,46]. In BeWo cell lines, B*a*P activates the AhR pathway with the induction of CYP1A1, increases syncytialization and the production of βhCG, and stabilizes p53 by phosphorylation with the induction of its target p21 [32]. However, it is important to highlight that, unlike trophoblastic cell lines, primary cytotrophoblasts are differentiated nonproliferative cells, which have exited the cell cycle. Thus, in primary VCT, the observed increase in p21 level cannot be correlated with cell cycle arrest. As pregnancy progresses, inner VCT fuse with the outer ST of the chorionic villi. This phenomenon is considered to indicate the terminal stage of CT differentiation and is mediated by the retroviral syncytin-1 protein [47]. The multinucleated ST layer of the placenta exhibits characteristics of senescent cells (including the expression of the senescence marker p53 and the CDK inhibitor p21, with a lack of proliferation) [48]. These senescence pathways are dysregulated up or down in placental pathologies, such as intrauterine growth restriction or pre-eclampsia [49,50]. The functional significance of senescence in ST for the human placenta is not understood. One hypothesis is that the resistance of senescent cells to apoptosis maintains ST viability throughout pregnancy, also involving upregulation of anti-apoptotic proteins of the Bcl-2 protein family, which is known to maintain senescent cell viability in other cell types [51]. The induction of p21 after B*a*P treatment and the decrease in Bcl-2 observed in the Cell Stress Array (Figure 4) could affect the syncytium and lead to premature aging of the placenta.

Western blot and RT-qPCR results indicate that the stabilization of p53 with the induction of its target p21 partially depends on the metabolism of B*a*P by the AhR pathway. Indeed, after treatment with B*a*P in the presence of the AhR antagonist, the stabilization of p53 and the induction of p21 were reduced, and no increase in γ-H2AX was observed. With the stabilization of p53, the induction of p21 and CYP1A1, and the phosphorylation of H2AX, we can assume that B*a*P preferentially induces genotoxicity with the absence of major oxidative stress at this concentration of B*a*P.

Overall, the Cell Stress Array highlights a greater number of actors whose protein levels decrease in co-exposure compared to individual exposure. This, therefore, suggests that other routes could be modulated in relation to individual exposure. However, the overall decreases observed after incubation with CeO_2_ NP could be due to their capacity to adsorb proteins on their surface [52] and form a corona. This could explain the differences in results observed between the Cell Stress Array, Western blot, and IHC. Indeed, for the Cell Stress Array and Western blots, protein extracts were centrifuged beforehand. The sedimented CeO_2_ NP may have carried proteins with them. However, the differences in IHC results may be due to the different response of all the villous cells compared to that of the trophoblasts only. In the cytoplasm, NP aggregates can sequestrate intracellular proteins. CeO_2_ NP could, thus, modulate the signaling pathways by capturing molecules that can no longer intervene in their pathway. B*a*P, being lipophilic and crossing plasma membranes freely, could facilitate the cellular internalization of NP. Conversely, the ability of nanoparticles to adsorb B*a*P on their surface could not only modify the internalization of B*a*P and CeO_2_ NP, but could also change the kinetics of B*a*P biotransformation in the cells and thus modify its toxicity. Additional studies are needed to verify these hypotheses.

## 4. Materials and Methods

### 4.1. Placenta Collection

The study was performed according to the principles of the Declaration of Helsinki. Placentas were collected from nonsmoking, healthy women with pregnancies delivered by caesarean section between 39 and 41 weeks of amenorrhea (hereafter called “term”). After obtaining written consent from informed patients and approval from our local ethics committee (CPP: 2015-May-13909), placentas were obtained from maternity hospitals in the Île-de-France region: Port-Royal Maternity, the Institut Mutualiste Montsouris, the Private Hospital of Antony, the Béclère Hospital, the Beaujon Hospital and the Diaconesses Hospital.

### 4.2. Cytotrophoblast Purification and Culture

After collection, placental tissues were washed in Ca^2+^- and Mg^2+^-free Hank’s balanced salt solution (HBSS, Gibco #14175, Thermo Fisher Scientific, Illkirch, France). Chorionic villi were gently scraped free from vessels and connective tissue, and dissected into approximate 25 mg fragments. The time for placental dissection was kept under 30 min to prevent tissue degradation. The mononucleated villous cytotrophoblasts were isolated, based on the methods of Kliman et al. [53]. After dissection, the chorionic villi were washed in Ca^2+^- and Mg^2+^-free HBSS, and then digested in trypsin digestion medium containing HBSS 5 mL/g, 0.1% trypsin (Gibco #27250-018), 0.1 M MgSO_4_ (Merck #5886-0500, New York, NY, USA), 0.1 M CaCl_2_ (Merck #1-02820-1000), 4% milk, and 50 Kunitz/mL DNAse type IV (Sigma-Aldrich #D5025, Saint Quentin Fallavier, France), for 30 min at 37 °C without stirring. The following digestions of 10 min, with renewal of the trypsin solution at each digestion, were monitored by observation under an optical microscope. Digestions containing a majority of VCT were kept and pooled. The chorionic villi were finally washed with warm HBSS (37 °C). Each time, the supernatant containing VCT was collected after tissue sedimentation, filtered (on 40 µm pore filters), and incubated with 10% fetal calf serum (FCS, vol/vol) to stop the trypsin activity. After purification by Percoll gradient, VCT were resuspended and cultured in Dulbecco′s modified Eagle′s medium (DMEM, containing 1 g/L glucose, pyruvate, without phenol red, Thermo Fisher Scientific, #11880, Illkirch, France) supplemented with 10% FCS (Eurobio #CVFSVF00-01, Les Ulis, France), 2 mM glutamine (Sigma-Aldrich #G7513), 100 IU/mL penicillin, and 100 µg/mL streptomycin (Gibco #15140-122, Thermo Fisher Scientific) at 150,000 cells/cm^2^ on 60 mm diameter culture dishes. After around 16 h of culture (here termed “overnight”), VCT were carefully washed to eliminate nonadherent cells. Purified VCT were characterized for each culture to ensure the homogeneity between purifications by microscopic visualization, the ability to aggregate at 24 h of culture and to form ST at 48 h, and by monitoring the production of hCG secreted into the supernatant.

### 4.3. Treatments

B*a*P (Sigma-Aldrich #B1760) was dissolved in DMSO at 10 mM and stored at 4 °C (stock solution). Trophoblasts were incubated with B*a*P for 30 min, 4 h, or 24 h at a final concentration of 0.1, 0.6, or 1 μM. CeO_2_ NP were obtained from the Joint Research Centre of the European Union (NM-212, IHCP, Ispra, Italy). The physical and chemical characterization of CeO_2_ NP had been previously performed [25]. NP were dispersed in DMEM cell culture media without FCS at a concentration of 3 mg/mL through sonication with a sonifier equipped with a cup horn (450 W and 50/60 Hz, Branson, Danbury, CT, USA) at 70% amplitude, on ice, for 2 min. NP were then sequentially diluted in cell culture medium with FCS immediately before use, to give a final concentration of 6.3 μg/cm^2^ (corresponding to 40 μg/mL as in our previous study [25]) or of 10 μg/cm^2^ (64 μg/mL). Trophoblasts were incubated with CeO_2_ NP for 30 min, 4 h, or 24 h. The AhR antagonist CH223191 (Sigma-Aldrich #182705) was used at 3 μM in culture medium (3 mM stock solution in DMSO) and p53 inhibitor pifithrin-α (Sigma-Aldrich #P4359) was used at 20 µM in culture medium (10 mM stock solution in DMSO). AhR and p53 inhibitors were incubated with cells for 1 h prior to B*a*P incubation. Staurosporin (STS, Sigma-Aldrich #S4400) was used at a final concentration of 0.5 µM for 2 to 6 h and TBHP (Luperox TBH70X solution, Sigma-Aldrich #458139) at 100 µM for 4 h.

### 4.4. Metabolic Activity by WST-1

Metabolic activity was assessed using the WST-1 assay (Sigma-Aldrich #11644807001)—in which mitochondrial dehydrogenase cleaves the tetrazolium salt 2-(4-iodophenyl)-3-(4-nitrophenyl)5-(2,4-disulfophenyl)-2H-tetrazolium (WST-1) into formazan—according to the manufacturer’s instructions. Trophoblast cells were cultured overnight in 48-well plates, washed with fresh medium and treated accordingly for 24 to 72 h. At the end of the exposure time, cells were rinsed with culture medium and then WST-1 reagent was added (1:100) to each well and incubated for 3 h at 37 °C. Spectrometric absorbance was measured using a microplate reader (EnSpire 2300 Multilabel reader, PerkinElmer, Villebon-sur-Yvette, France) at 440 nm, and using 600 nm as correction wavelength (to remove the nonspecific emission). To measure the background noise from NP, the same experiment was performed in parallel, followed by the addition of Triton to each well 15 min before adding WST-1 reagent, and the corresponding absorbances were subtracted from the results.

### 4.5. Protein Extraction and Western Blot

Total protein extracts from trophoblast cells from 60 mm dish cultures were obtained by harvesting cells in Laemmli lysis buffer, with added protease inhibitors (1:100, Protease Inhibitor Cocktail Set I, Calbiochem Merck #539131) and phosphatase inhibitors (1:50, Phosphatase Inhibitor Cocktail Set V, Calbiochem Merck # 524629). Protein extracts were centrifuged for 5 min at 14,000× *g* and stored at −80 °C. Protein concentrations were determined using the Pierce™ BCA Protein Assay Kit (Thermo Fisher Scientific #23235). Equal amounts of proteins (30 µg) were separated on 7.5%, 4–15%, or 8–16% SDS-PAGE mini-PROTEAN^®^ TGX™ precast protein gel (Bio-Rad #4561085 #4561084 #4561103, Marnes-la-Coquette, France) under reducing conditions (dithiothreitol 10%, Sample Reducing Agent 10×, Invitrogen #NP0009, Waltham, MA, USA) and transferred onto a nitrocellulose membrane (Trans-blot Turbo Transfer pack, 0.2 µm nitrocellulose, Bio-Rad #1704159). Blots were incubated overnight with the primary antibody at 4 °C, and then for 1 h with the appropriate DyLight 680 or 800 Fluor-conjugated secondary antibody (Thermo Fisher Scientific #35568 and #35521). The primary antibodies used were mouse anti-AhR (1:500, Sigma-Aldrich #WH0000196-M2), rabbit anti-CYP1A1 (1:1000, Proteintech #3241-1-AP, Rosemont, IL, USA), rabbit anti-HO-1 (1:1000, Enzo Life Sciences #ADI-SPA-895D, Farmingdale, NY, USA), rabbit anti-ɣ-H2AX (1:1000, Cell Signaling, #9718, Danvers, MA, USA), rabbit anti-NFκB p65 (1:2000, Abcam #ab16502, Cambridge, UK), rabbit anti-p21 (1:1000, Cell Signaling, #2947), mouse anti-p53 (1:1000, Cell Signaling #18032), rabbit anti-PARP-1 (1:1000, Cell Signaling #9542), rabbit anti-Prx-SO_3_ (1:500, Abcam #ab16830), mouse anti-vinculin (1:1000, Sigma-Aldrich 1V9131). Secondary antibodies were DyLight 800-labeled anti-mouse (1:20,000, Thermo Fisher Scientific #35521) or DyLight 680-labeled anti-rabbit (1:20,000, Thermo Fisher Scientific #35568). Primary and secondary antibodies were diluted in the following dilution buffer: 1X TBS, 0.1% Tween-20, 5% nonfat dry milk, 0.1% sodium azide. Blots were scanned with an Odyssey^®^ Imaging System (Li-COR, Bad Homburg, Germany). Quantification was performed using the Li-COR Odyssey software.

### 4.6. hCG ELISA

The hCG assay was carried out in the centrifuged supernatants (3 min at 500× *g*) of the VCT exposed to pollutants for 48 h. The assays were performed according to the supplier’s instructions (Abcam^®^ ab100533) by sandwich ELISA (enzyme-linked immunosorbent assay). The calibration range was established with recombinant human hCG and varying from 54.87 pg/mL to 40,000 pg/mL. The range and the samples were deposited in duplicate. The absorbance was measured at 450 nm using a microplate reader (EnSpire 2300 Multilabel reader, PerkinElmer™) from which the absorbance measured at 600 nm was removed in order to exclude nonspecific emissions from the plate. These results were normalized to the quantity of protein present in each culture well.

### 4.7. Cell Stress Array

The effects of B*a*P and CeO_2_ NP on signaling involved in cellular stress were evaluated using the Human Cell Stress Array kit (R&D systems #ARY018) in accordance with the manufacturer’s instructions. Total protein extracts obtained with the lysis buffer from the kit were incubated for 30 min on a rocking platform shaker at 4 °C. Equal amounts of each total protein extract (*n* = 5) for each condition were pooled to obtain 300 μg of a representative sample for each condition. Membranes were saturated for 1 h with the buffer provided and then incubated overnight at 4 °C with the protein extracts and the detection antibody cocktail. Membranes were washed three times for 10 min and incubated with streptavidin coupled with a fluorochrome (1:2000, LI-COR Biosciences IRDye #926-32230). Blots were scanned with an Odyssey^®^ Imaging System (Li-COR, Bad Homburg, Germany). Quantification was performed using the Li-COR Odyssey software.

### 4.8. RNA Extraction, Reverse Transcription, and Real-Time Quantitative PCR

The treated VCT were recovered by trypsin-EDTA 0.05% (Gibco™ 25300054). Total RNA extraction was performed on pellets of 3 million cells with the RNeasy^®^ mini kit (Qiagen^®^ 74104) according to the supplier’s instructions, with DNase (Qiagen^®^ 79254). After elution of the RNAs, the RNA concentration (A260) and the purity of the samples (A260/A280 and A260/A230 ratios) were determined by a spectrophotometer (Nanodrop ND-1000). RNAs with an A260/A230 ratio of less than 1.6 were purified with the Qiagen^®^ RNeasy^®^ MinElute Cleanup kit (74204). The total RNAs recovered were stored at −80 °C after addition of 1 µL of RNase-OUT (Invitrogen™ 10777019). Reverse transcription was performed using the Transcriptase inverse SuperScript™ III kit (Invitrogen™ 18080-044) using random primer (Invitrogen, Paisley, UK) for 1 µg RNA. RT-qPCR was carried out in a 10 µL reaction volume containing 10 ng of cDNA, 5 µM of each primer, and 5 µL of SYBR™ Green (Master Mix PowerTrack™ Applied Biosystems™ A46111) using the CFX384 Real-Time PCR Detection System (Bio-Rad™). All primers (Table 1) were validated beforehand [5]. PCR cycles consisted of the following steps: polymerase activation (2 min, 95 °C), denaturation (15 s, 95 °C), and annealing and extension (1 min, 60 °C), followed by a melting curve to ensure no contamination. The threshold cycle (Ct) was measured as the number of cycles at which the fluorescence emission first exceeds the background. The relative amounts of mRNA were estimated using the ∆∆Ct method and then expressed as fold change [54]. Primers for the reference genes HPRT, RPLO, and SDHA were used for the normalization of the results. Each gene was normalized to the geometric mean of the 3 reference genes.

### 4.9. Immunochemistry

Placental explants treated for 24 h with DMSO, 1 µM of B*a*P, and/or 10 µg/cm^2^ of CeO_2_ NP (*n* = 3 per condition) were fixed in 4% paraformaldehyde (Electron Microscopy Sciences, Hatfield, PA, USA) overnight at 4 °C, and then dehydrated and embedded in Paraplast^®^. Placental tissue sections (5 µm) were deparaffinized, rehydrated in ethanol/water, and processed for heat-induced antigen retrieval in citrate buffer, pH 6.1 or Tris/EDTA buffer pH 9 (Dako, Trappes, France) for 45 min at 90 °C in a water bath. Sections were permeabilized in 0.3% Triton X-100 for 4 min, and then incubated with rabbit anti-hypoxia-inducible factor 1α (HIF-1α) antibody diluted 1:100 (200 µg/mL; Novus Biologicals #NB100-449, Englewood, CO, USA), rabbit anti-hypoxia-inducible factor 2α (HIF-2α) antibody diluted 1:500 (1 mg/mL; Novus Biologicals #NB100-122), rabbit anti-p21 antibody diluted 1:50 (244 µg/mL; Cell Signaling #2947), mouse anti-p53 antibody diluted 1:100 (237 µg/mL; Dako #M7001), or nonspecific rabbit/mouse IgG as controls. Staining was visualized using Novolink™ Polymer Detection Systems (Leica Biosystems). Slides were scanned using a Lamina multilabel slide scanner (Perkin Elmer) using brightfield imaging at a magnification of ×20. Quantification of protein expression was performed on ten random 100 × 100 pixel images per sample (*n* = 3 per condition, resulting in a total of 30 images) using the ImageJ 1.52 software, and thresholding analysis was performed as previously described [55].

### 4.10. Statistical Analysis

Statistical tests were performed using the GraphPad Prism 7.04 software (La Jolla, CA, USA). The results of the quantitative analysis are presented as means ± standard error of the mean (SEM). The Shapiro–Wilks normality test was performed to determine whether the samples followed a normal distribution. Differences between groups were evaluated with the parametric paired Student’s *t*-test for samples with a normal distribution and with the nonparametric Wilcoxon–Mann–Whitney test for samples with a non-normal distribution. A *p*-value lower than 0.05 was considered to be statistically significant, with *p* < 0.05, *p* < 0.01, *p* < 0.001 *p* < 0.0001 represented as *, **, *** and ****, respectively.

## 5. Conclusions

Few data are available concerning the adverse effects of women’s exposure to cerium dioxide nanoparticles in the sensitive window of human pregnancy, and even fewer findings have been reported concerning the effects of their co-exposure with other pollutants, such as B*a*P. Exploring the toxicological consequences of placental barrier exposure to mixtures of pollutants, which are representative of environmental exposure, and identifying which cell signaling pathways are impaired should provide insights concerning placental toxicology, with potent implications for pregnancy outcomes. In conclusion, in this study, we identified the variations in major cellular actors in xenobiotic detoxification (AhR, CYP1A1), cell stress and genotoxicity (p53, p21, γ-H2AX), hypoxia response (HIF), and oxidative stress response (Prx-SO_3_) after VCT incubation with CeO_2_ NP and B*a*P individually and in combination. These results provide new perspectives for the study of the mechanisms by which these two very common air pollutants impair placental development and function in pregnant women.

## Figures and Tables

**Figure 1 ijms-24-05439-f001:**
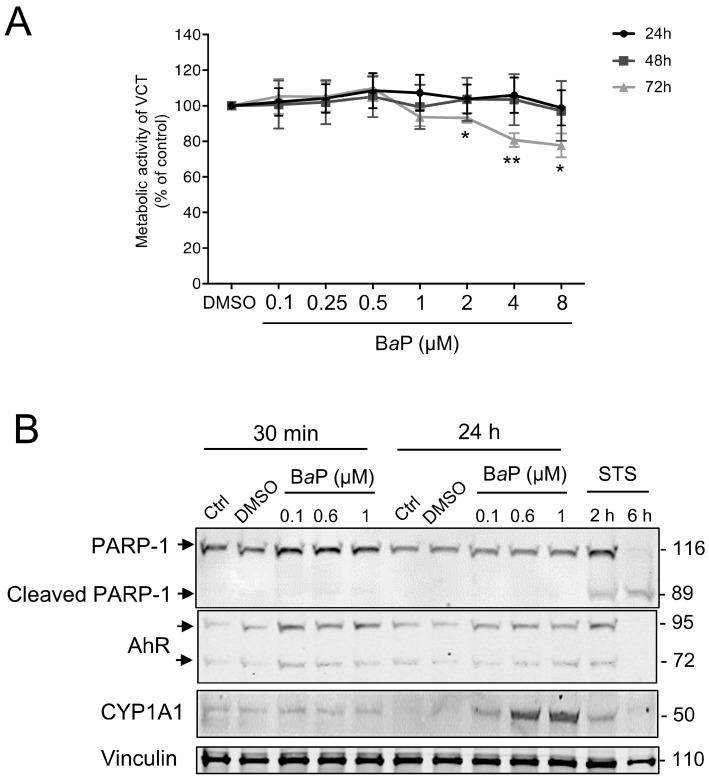
Effect of increasing concentrations of benzo(a)pyrene on human primary trophoblast metabolic activity and AhR pathway activation. Villous cytotrophoblasts (VCT) purified from term placentas were plated overnight. They were then either untreated (Ctrl) or incubated with dimethyl sulfoxide (DMSO, solvent control), benzo(a)pyrene (B*a*P), or cerium dioxide nanoparticles (CeO_2_ NP) at the indicated concentrations for 30 min, 24 h, 48 h, or 72 h. (**A**) WST-1 assay showing metabolic activity of VCT as mean ± SEM (*n* = 3) relative to DMSO control for each time of treatment. * *p* < 0.05, ** *p* < 0.01 vs. DMSO. (**B**) VCT were also incubated with 0.5 µM of staurosporine (STS) for 2 h or 6 h. Total protein extracts were subjected to SDS-PAGE under reducing conditions and membranes were immunoblotted with anti-PARP-1, anti-AhR, anti-CYP1A1, and anti-vinculin antibodies (the latter was used as loading control).

**Figure 2 ijms-24-05439-f002:**
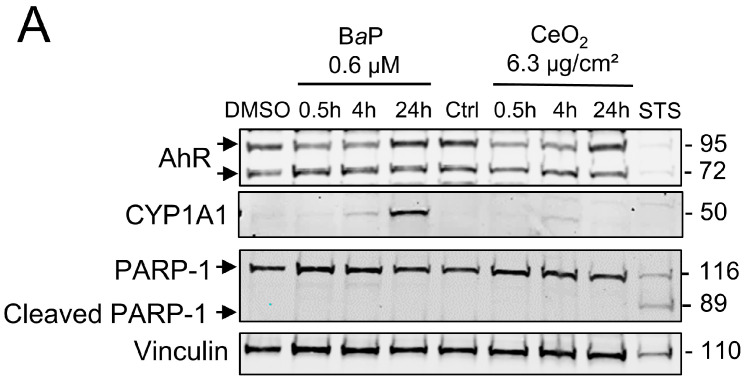
Effect of benzo(a)pyrene and cerium dioxide nanoparticles on the AhR pathway and metabolic activity of human primary trophoblasts. VCT purified from term placentas were plated overnight. They were then incubated either with DMSO or with B*a*P and/or CeO_2_ NP at the indicated concentrations for 30 min, 4 h, 24 h, or 72 h, or with 0.5 µM of STS for 4 h. (**A**) Total protein extracts were subjected to SDS-PAGE under reducing conditions and membranes were immunoblotted with anti-AhR, anti-CYP1A1, anti-PARP-1, and anti-vinculin antibodies (the latter was used as loading control). (**B**) WST-1 assay showing metabolic activity of VCT as mean ± SEM (*n* = 3) relative to DMSO after 72 h of treatment. * *p* < 0.05, ** *p* < 0.01 vs. DMSO. (**C**) The human chorionic gonadotropin (hCG) assay was performed by enzyme-linked immunosorbent assay in VCT supernatants. VCT were incubated for 48 h with 1 µM of B*a*P and/or 10 μg/cm^2^ of CeO_2_ NP, or 500 µM of TBHP. The hCG assays were normalized to µg of protein per condition. Results represent the mean ± SEM (*n* = 3) relative to control (Ctrl). **** *p* < 0.0001 vs. Ctrl.

**Figure 3 ijms-24-05439-f003:**
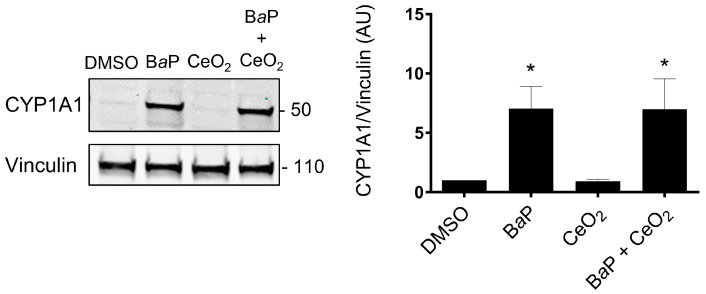
Homogeneity of the cellular response to benzo(a)pyrene and cerium dioxide nanoparticles in independent VCT cultures from 5 placentas. VCT purified from 5 different term placentas were plated overnight in culture and then were incubated either with DMSO or with B*a*P (0.6 µM) or CeO_2_ NP (6.3 µg/cm^2^), and in B*a*P and CeO_2_ NP co-exposure for 24 h. Total protein extracts (30 µg) were subjected to SDS-PAGE under reducing conditions and membranes were immunoblotted with anti-CYP1A1 and anti-vinculin (loading control) antibodies. Immunoblots were quantified with an Odyssey System Imager and results in the bar scale graph represent the mean ± SEM after normalization to vinculin (*n* = 5). * *p* < 0.05 vs. DMSO.

**Figure 4 ijms-24-05439-f004:**
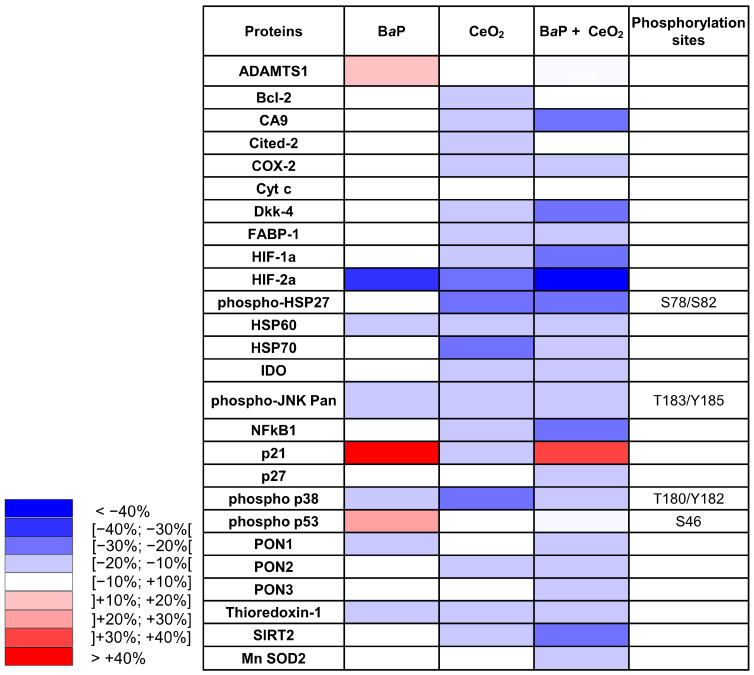
Cell Stress Array heatmap of the relative variations in protein expression after exposure of human primary trophoblasts to benzo(a)pyrene and cerium dioxide nanoparticles. VCT purified from term placentas were plated overnight and were either untreated or incubated with B*a*P (0.6 µM) or CeO_2_ NP (6.3 µg/cm^2^) alone or together for 24 h. Total protein extracts were pooled (*n* = 5) and protein expression was measured by the Cell Stress Array. Immunoblots were quantified with an Odyssey System Imager and the results are shown in the heatmap as variation relative to the DMSO-treated cells.

**Figure 5 ijms-24-05439-f005:**
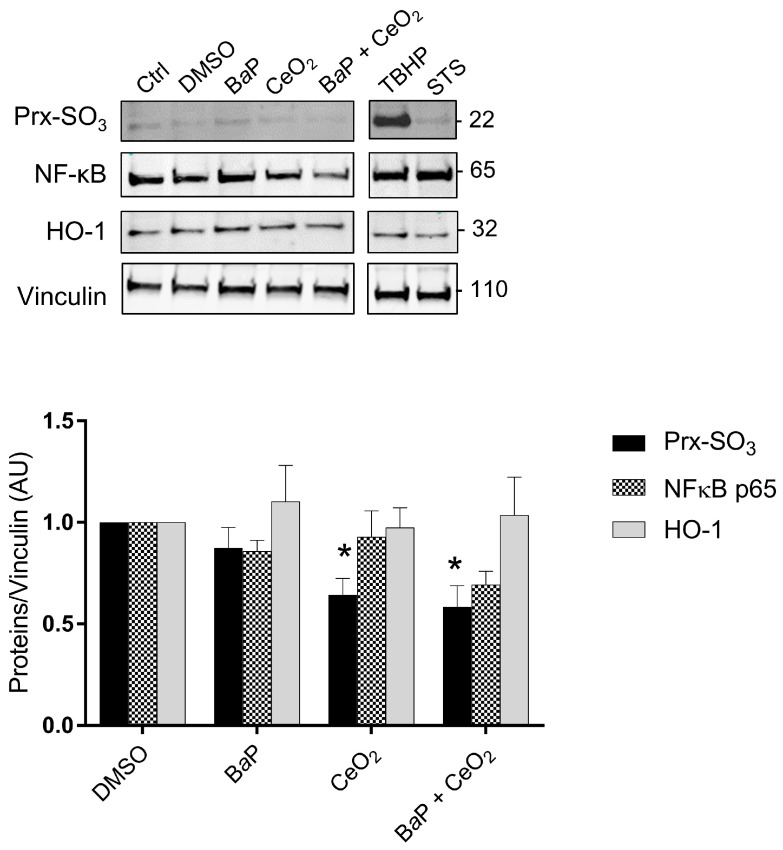
Intracellular oxidative stress after trophoblast exposure to benzo(a)pyrene and/or cerium dioxide nanoparticles. VCT purified from term placentas were plated overnight and were untreated (control) or incubated with DMSO, B*a*P (0.6 µM), or CeO_2_ NP (6.3 µg/cm^2^) and in co-exposure for 24 h. *Tert*-butyl hydroperoxide (TBHP, 100 μM, 4 h) and STS (0.5 μM, 3 h) were used as positive controls for oxidative stress and apoptosis, respectively. Total protein extracts were subjected to SDS-PAGE under reducing conditions and membranes were immunoblotted with anti-Prx-SO_3_, anti-NFκB p65, anti-HO-1, and anti-vinculin antibodies (loading control). Immunoblots were quantified with an Odyssey System Imager and the results represent the mean ± SEM after normalization to vinculin (*n* = 5). * *p* < 0.05 vs. DMSO.

**Figure 6 ijms-24-05439-f006:**
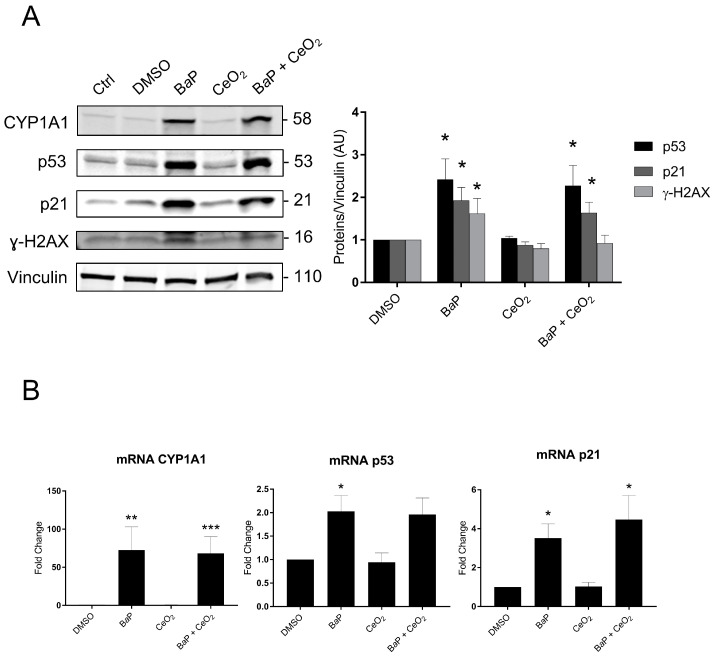
p53 signaling and DNA damage after trophoblast exposure to benzo(a)pyrene and/or cerium dioxide nanoparticles. VCT purified from term placentas were plated overnight and were untreated (control) or incubated with DMSO, B*a*P (1 µM), and/or CeO_2_ NP (10 µg/cm^2^) for 24 h. (**A**) Total protein extracts were subjected to SDS-PAGE under reducing conditions and membranes were immunoblotted with anti-CYP1A1, anti-p53, anti-p21, anti-γH2AX, and anti-vinculin antibodies (loading control). Immunoblots were quantified with an Odyssey System Imager and the results represent the mean ± SEM after normalization to vinculin (*n* = 6). * *p* < 0.05 vs. DMSO. (**B**) Total mRNA was extracted from VCT, followed by reverse transcription. Real-time quantitative PCR (RT-qPCR) was performed for the genes of interest CYP1A1, p53, and p21. The results, expressed as fold change, represent the mean ± SEM after normalization to the geometric mean of the reference genes HPRT, RPLO, and SDHA (*n* = 5). * *p* < 0.05, ** *p* < 0.01, *** *p* < 0.001 vs. DMSO.

**Figure 7 ijms-24-05439-f007:**
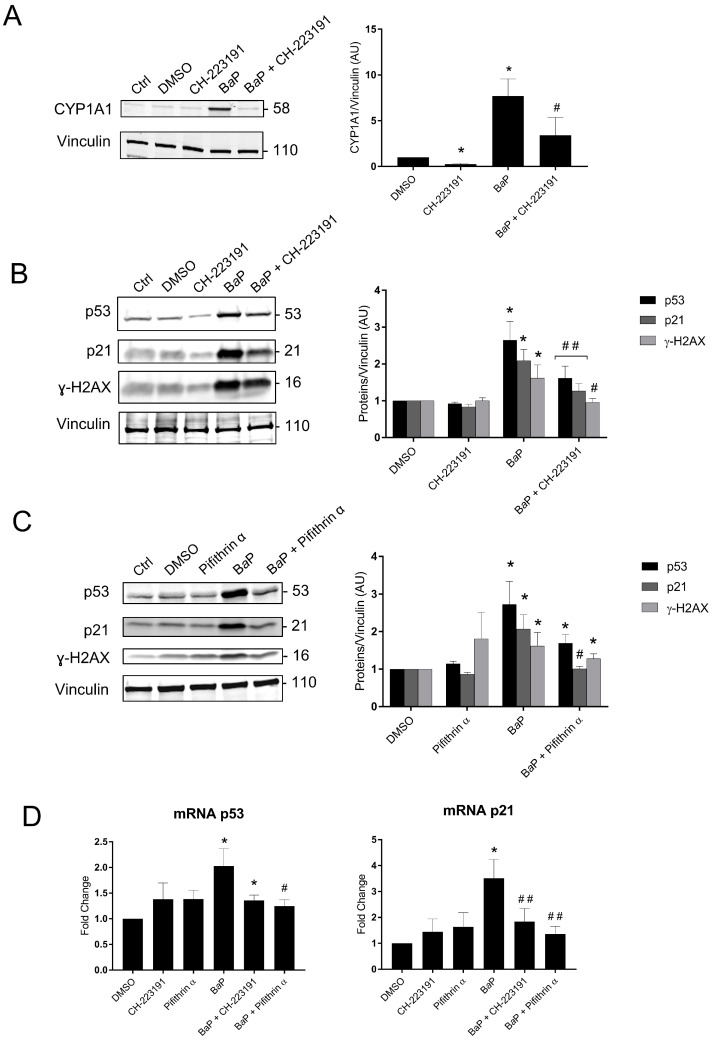
Protein expression and mRNA levels of p53 signaling and the AhR pathway after trophoblast exposure to benzo(a)pyrene. (**A**–**C**) VCT purified from term placentas were plated overnight and were untreated (control), incubated with DMSO or B*a*P (1 µM), preincubated for 1 h with AhR antagonist (3 µM), preincubated for 1 h with pifithrin-α (20 µM), incubated with B*a*P (1 µM) after preincubation for 1 h with AhR antagonist for 24 h, or incubated with B*a*P (1 µM) after preincubation for 1 h with pifithrin-α for 24 h. Total protein extracts were subjected to SDS-PAGE under reducing conditions and membranes were immunoblotted with anti-CYP1A1, anti-p53, anti-p21, anti-γH2AX, and anti-vinculin antibodies (loading control). Immunoblots were quantified with an Odyssey System Imager and the results represent the mean ± SEM after normalization to vinculin (*n* = 6). * *p* < 0.05 vs. DMSO, # *p* < 0.05 vs. B*a*P, ## *p* < 0.01 vs. B*a*P. (**D**) VCT purified from term placentas were plated overnight and incubated with DMSO or B*a*P (1 µM), preincubated for 1 h with AhR antagonist (3 µM) or pifithrin-α (20 µM), or incubated with B*a*P (1 µM) after preincubation for 1 h with AhR antagonist or pifithrin-α for 24 h. Total mRNA was extracted from VCT, followed by reverse transcription. RT-qPCR was performed for the genes of interest, i.e., p53 and p21. The results, expressed as fold change, represent the mean ± SEM after normalization to the geometric mean of the reference genes HPRT, RPLO, and SDHA (*n* = 5). * *p* < 0.05 vs. DMSO, # *p* < 0.05 vs. B*a*P, ## *p* < 0.01 vs. B*a*P.

**Figure 8 ijms-24-05439-f008:**
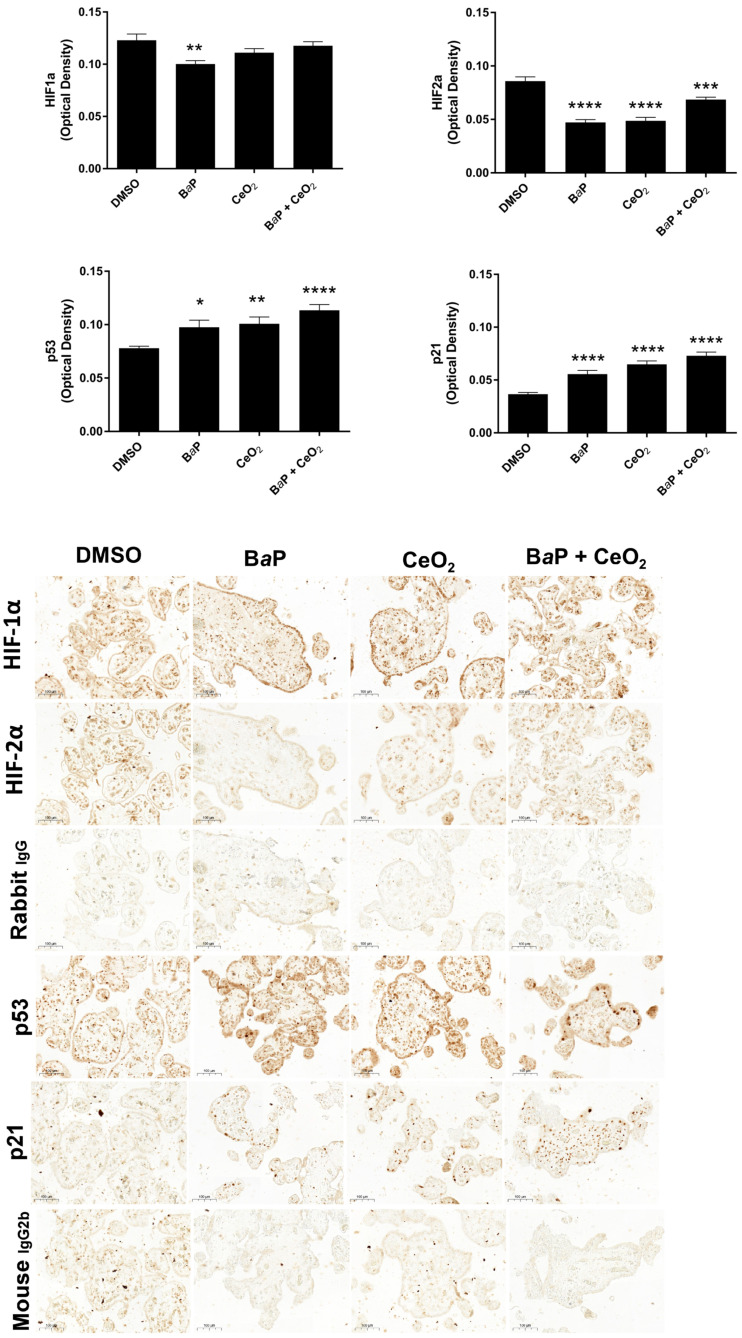
Immunohistochemistry of HIF, p53, and p21 after chorionic villi exposure to benzo(a)pyrene or/and cerium dioxide nanoparticles. Representative images of immunohistochemical staining of HIF-1α, HIF-2α, p53, p21, and negative controls (rabbit IgG and mouse IgG2b) from placental explants incubated with DMSO, B*a*P (1 µM), and/or CeO_2_ NP (10 µg/cm^2^) for 24 h (*n* = 3). Quantitative analysis of HIF-1α, HIF-2α, p53, and p21 expression was performed using the ImageJ plugin. Values represent means ± SEM. * *p* < 0.05, ** *p* < 0.01, *** *p* < 0.001, **** *p* < 0.0001 vs. DMSO.

**Table 1 ijms-24-05439-t001:** Characteristics of the primers used.

Genes		Primer Sequences	Efficiency (%)	Tm (°C)	Fragment Length
CYP1A1	F	5′-CCA-CAG-CAC-AAC-AAG-AGA-C-3′	98.03	54	150
	R	5′-CCA-TCA-GGG-GTG-AGA-AAC-3′		54	
p53	F	5′-AGG-CCT-TGG-AAC-TCA-AGG-AT-3′	94.86	58	141
	R	5′-CTG-AGT-CAG-GCC-CTT-CTG-TC-3′		59	
p21	F	5-CGT-GAG-CGA-TGG-AAC-TTC-3′	99.71	54	373
	R	5′-TCC-TGT-GGG-CGG-ATT-AG-3′		55	
RPLO	F	5′-AAC-ATC-TCC-CCC-TTC-TCC-T-3′	95.84	56	209
	R	5′-ACT-CGT-TTG-TAC-CCG-TTG-AT-3′		57	
HPRT	F	5′-GGC-GTC-GTG-ATT-AGT-GAT-G-3′	93.68	56	179
	R	5′-CAG-AGG-GCT-ACA-ATG-TGA-TG-3′		55	
SDHA	F	5′-TGG-GAA-CAA-GAG-GGC-ATC-TG-3′	98.75	59	86
	R	5′-CCA-CCA-CTG-CAT-CAA-ATT-CAT-G-3′		58	

## Data Availability

Data available on request from the authors.

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
