# Peer review of "Benzo(a)pyrene and Cerium Dioxide Nanoparticles in Co-Exposure Impair Human Trophoblast Cell Stress Signaling"

_ijms, 2023, doi:10.3390/ijms24065439_

Round 1

Reviewer 1 Report

The manuscript by Deval et al. describes the effects of coexposure to Benzo a pyrene (BaP) and CeO nanoparticles on cytotrophoblasts.

This paper is a series of previously published papers demonstrating the toxicity of individual exposure of CeO NPs on human trophoblasts  and BaP. The paper is written in a logical and understandable manner. The manuscript ay be of interest to IJMS readers. The corresponding nanoparticles used in this study have been extensively described and physiochemically characterized. I would recommend this study for publication in the International Journal of Molecular Sciences after making minor revisions based on the comments provided in this review.

It is not quite clear how the authors decided to choose the co-exposure concentrations. Why they used the ratio of 1 µM BaP per 10 µg/cm2 ?  Is it is assumed that for the ratio of 15 to 2 mg PAH per g UFP all UFP are indeed only CeO NP? Due to its lipophilic nature benzo pyrene is likely to diffuse from the inhalation site into the blood circulation. However, for CeO NPs the amount of NP that it can be effectively translocated from the lung into the circulation is very low, therefore, circulating concentrations would be substantially lower than ambient air concentrations..

If in the previous study by Nedder 2020 the authors demonstrated that CeO NPs are able to induce cytotoxicity and physiological effects on human trophoblasts (decreased the fusion ability), why did the authors not decide to explore whether co-exposure with benzo pyrene could have synergistic or additive effects on these parameters?

Line 594. I think this hypothesis is interesting and easy to test. The nanoparticle precipitates can be boiled with Laemmli buffer to denature/detach the adsorbed proteins. Then, samples can be and run on SDS-PAGE to determine if the NPs adsorb the proteins of interest.

Reviewer 2 Report

In this manuscript the authors described the main signaling pathways modulated after exposure to BaP or CeO2 NP on the human placenta.

The topic is very interesting and of high significance.

The structure of the article is well-organized. The materials and methods chapter were described in great detail. The authors presented essential figures. The figures are legible and well-described. The stages of research and results obtained were logically presented. 

The data looks very good and ready for publication.

Author Response

We thank the reviewer for the critical consideration of our work. We are very pleased with the reviewer's positive opinion of our manuscript.